# The Role of Early Rehabilitation and Functional Electrical Stimulation in Rehabilitation for Cats with Partial Traumatic Brachial Plexus Injury: A Pilot Study on Domestic Cats in Portugal

**DOI:** 10.3390/ani14020323

**Published:** 2024-01-20

**Authors:** Débora Gouveia, Ana Cardoso, Carla Carvalho, Inês Rijo, António Almeida, Óscar Gamboa, Bruna Lopes, Patrícia Sousa, André Coelho, Maria Manuel Balça, António J. Salgado, Rui Alvites, Artur Severo P. Varejão, Ana Colette Maurício, António Ferreira, Ângela Martins

**Affiliations:** 1Arrábida Veterinary Hospital—Arrábida Animal Rehabilitation Center, 2925-538 Setúbal, Portugal; p60855@ipluso.pt (D.G.); anacardosocatarina@gmail.com (A.C.); mv.carla.c@gmail.com (C.C.); inesrr@live.com.pt (I.R.); vetarrabida.lda@gmail.com (Â.M.); 2Superior School of Health, Protection and Animal Welfare, Polytechnic Institute of Lusophony, Campo Grande, 1950-396 Lisboa, Portugal; 3Faculty of Veterinary Medicine, Lusófona University, Campo Grande, 1749-024 Lisboa, Portugal; 4Faculty of Veterinary Medicine, University of Lisbon, 1300-477 Lisboa, Portugal; antonioalmeida@fmv.ulisboa.pt (A.A.); ogamboa@fmv.ulisboa.pt (Ó.G.); aferreira@fmv.ulisboa.pt (A.F.); 5Departamento de Clínicas Veterinárias, Instituto de Ciências Biomédicas de Abel Salazar (ICBAS), Universidade do Porto (UP), Rua de Jorge Viterbo Ferreira, n° 228, 4050-313 Porto, Portugal; brunisabel95@gmail.com (B.L.); pfrfs_10@hotmail.com (P.S.); andrefmc17@gmail.com (A.C.); mariamanuel.balca@gmail.com (M.M.B.); ruialvites@hotmail.com (R.A.); 6Centro de Estudos de Ciência Animal (CECA), Instituto de Ciências, Tecnologias e Agroambiente da Universidade do Porto (ICETA), Rua D. Manuel II, 4051-401 Porto, Portugal; 7Associate Laboratory for Animal and Veterinary Science (AL4AnimalS), 1300-477 Lisboa, Portugal; avarejao@utad.pt; 8Life and Health Sciences Research Institute (ICVS), School of Medicine, University of Minho, Campus de Gualtar, 4710-057 Braga, Portugal; asalgado@med.uminho.pt; 9ICVS/3B’s—PT Government Associate Laboratory, 4710-057 Braga, Portugal; 10Instituto Universitário de Ciências da Saúde (CESPU), Avenida Central de Gandra 1317, 4585-116 Gandra, Portugal; 11Department of Veterinary Sciences, Universidade de Trás-os-Montes e Alto Douro (UTAD), Quinta de Prados, 5000-801 Vila Real, Portugal; 12Centro de Ciência Animal e Veterinária (CECAV), Universidade de Trás-os-Montes e Alto Douro (UTAD), Quinta de Prados, 5001-801 Vila Real, Portugal; 13CIISA—Centro Interdisciplinar-Investigação em Saúde Animal, Faculdade de Medicina Veterinária, Av. Universidade Técnica de Lisboa, 1300-477 Lisboa, Portugal

**Keywords:** brachial plexus injury, cats, electrical stimulation, locomotor training, rehabilitation

## Abstract

**Simple Summary:**

Partial traumatic brachial plexus injury due to road traffic accidents is one of the most common and challenging disorders requiring neurorehabilitation in cats. The implementation of early intensive neurorehabilitation, including electrical stimulation with specific parameters and physical activity, may contribute to faster sensory-motor recovery and resumption of ambulation, possibly avoiding amputation of the affected limb.

**Abstract:**

This prospective observational cohort pilot study included 22 cats diagnosed with partial traumatic brachial plexus injury (PTBPI), aiming to explore responses to an early intensive neurorehabilitation protocol in a clinical setting. This protocol included functional electrical stimulation (FES), locomotor treadmill training and kinesiotherapy exercises, starting at the time with highest probability of nerve repair. The synergetic benefits of this multimodal approach were based on the potential structural and protective role of proteins and the release of neurotrophic factors. Furthermore, FES was parametrized according to the presence or absence of deep pain. Following treatment, 72.6% of the cats achieved ambulation: 9 cats within 15 days, 2 cats within 30 days and 5 cats within 60 days. During the four-year follow-up, there was evidence of improvement in both muscle mass and muscle weakness, in addition to the disappearance of neuropathic pain. Notably, after the 60 days of neurorehabilitation, 3 cats showed improved ambulation after arthrodesis of the carpus. Thus, early rehabilitation, with FES applied in the first weeks after injury and accurate parametrization according to the presence or absence of deep pain, may help in functional recovery and ambulation, reducing the probability of amputation.

## 1. Introduction

The brachial plexus is a complex anatomic structure that is formed from a network of ventral roots of the spinal cord segments C6 to T2; in cats, it is located between the intervertebral foramen and the thoracic limb [1].

Partial traumatic brachial plexus injury (PTBPI) is one of the most common peripheric nervous system lesions that affects the brachial plexus in cats [2,3,4], most commonly due to road traffic accidents [5]. This type of lesion may occur due to excessive traction of the thoracic limb or severe abduction of the scapula [5,6,7].

The diagnosis of PTBPI is usually based on history, clinical signs and findings of both clinical and neurological examinations [2,6]. However, electrodiagnostic assessment can be useful as a precise tool for identifying damaged nerves through the resulting compound muscle action potentials (CMAPs), which are a reflection of muscle force in normal and re-innervated muscles [8]. These tests have already been used in cats and may be sufficient to localize injury of the brachial plexus, but they are still not sufficient to pinpoint the exact location of specific injured nerves [9].

Previous studies based on nerve conduction and electromyograms (EMG), which are not invasive, showed that these tools may have a delayed diagnostic role but are not best used immediately after injury, being more effective, for example, six weeks later, when fibrillations in the de-innervated muscles occur [10,11,12]. In addition, magnetic resonance imaging has been described as a possible complementary exam to identify brachial plexus masses [9].

Seddon (1943) [13] was the first to implement a classification system for nerve injuries, with three categories based on the presence of demyelination and the extent of damage to the axons and/or the connective tissues around the nerve [14,15]. Thus, since 1947, nerve injuries have been classified as follows: neuropraxia, which involves focal demyelination without damage to the axons and/or connective tissues and is mostly due to mild compression or nerve traction, leading to a transitory interruption or decrease in velocity conduction and muscle weakness, with recovery possible in three to six weeks; axonotmesis (crush injury), which involves direct damage to the axons and focal demyelination, while maintaining the continuity of nervous connective tissue, with recovery possible in 12–18 months; and neurotmesis, which involves a full transection of the axons and connective tissue [10,13,14,15,16,17]. 

In human medicine, a more aggressive approach, with acute repair, leads to better functional recovery; the preference is to avoid the “wait and see” approach, which can be much more costly [10,18]. Additionally, early nerve repair may result in improved functional outcomes, with less muscle fibrosis and secondary atrophy, as atrophy begins soon after denervation. However, the success rate may depend on various factors, such as the time between injury and surgery, the distance from the injury site to the target muscle, location of the injury (i.e., nerve root avulsion) and the regeneration rate. Thus, the time needed for re-innervation of sensory receptors is much longer than that needed for motor nerves, but better outcomes can be achieved with early repair [10]. 

PTBPI is mostly seen in cats, particularly as a result of road traffic collisions, when an impact force promotes excessive medial or caudal movement of the limb. Lesions between C8–T1 are the most common; however, C6–C7 can also be involved [5].

Clinical signs include a gait alteration compatible with proximal radial nerve paralysis, inability to support weight and inability to extend the carpus/digits, resulting in pain and dragging of the limb on the floor [3,19]. In addition, the elbow can be ventrally positioned, resulting in a dropped elbow posture. This change in positioning is due to shoulder plexus paralysis, specifically, paralysis of the latissimus dorsi and triceps long head, which are innervated by the caudal roots of the plexus [1,19,20]. 

Avulsion of the C8 rootlets affects the cutaneous trunci reflex on that side, regardless of the intensity of sensory stimulation. If the cranial roots of the plexus are preserved (C6-C7), with the musculocutaneous nerve intact, the elbow stays in a flexed position, resulting in a lameness score of five [1,3,19,20].

Cases of total brachial plexus avulsion may result in an absence of superficial pain distal to the elbow and the lateral surface of the limb. However, if the cranial portion of the brachial plexus is preserved, this pain can be absent only distal to the elbow, with sensation on the medial side of the thoracic limb intact [1,3,20]. 

Electrical stimulation (ES) is a treatment modality for peripheral nerve injury that can allow reinnervation of affected muscles and enhance functional outcomes. This treatment modality is well accepted in cats [21], and outcomes can be improved by daily stimulation. The underlying mechanism involves upregulation of intramuscular neurotrophic factors, which may be essential for long-term changes in the functional muscle. An understanding of this mechanism has already changed the paradigm in rats [22,23,24].

A therapeutic approach based on ES aims to improve post-traumatic neuromuscular recovery and can be applied either to the damaged nerve [25,26,27] or the denervated muscles [28,29,30], as was already tried in rat models of axonotmesis [25,31,32,33]. The degree of tissue healing is highly variable [20,34,35], but early neurorehabilitation protocols may help to improve the outcome. 

In human medicine, neurorehabilitation protocols for treatment of denervated muscles based on single sessions of ES in combination with physical exercise, muscle stretching and passive movements lead to enhanced contraction without muscle fatigue [25,29]. 

Research in animal models (i.e., rats), also revealed that applying low-intensity ES directly to the nerve through implanted electrodes increased functional and morphological regeneration of the nerve, possibly via a delay in axonal degeneration, stimulation of nerve sprouting and regeneration of the myelin sheath [23,36]. Thus, low-frequency stimulation can increase the number of myelinated fibers and the axon density and result in a higher ratio of blood vessels to total nerve area compared to the values seen in non-stimulated and injured nerves. Furthermore, other studies have shown that ES is a strong delayer of degeneration in denervated muscle [25,37] and regulates molecular alterations in skeletal muscles [24,29]. 

The main effects of ES are still under investigation, and there is a need to elucidate how different frequencies may affect the regeneration of nerve fibers.

Treatment of PTBPI can be conservative, surgical or mixed. Conservative management can be based on functional neurorehabilitation and prevention of the self-mutilation and secondary wounds that can be associated with carpal arthrodesis, with the aim of preventing contracture and loss of carpus functionality [7,38,39,40]. 

The first aim of this study was to describe recovery from PTBPI in cats after implementation of a mixed multidisciplinary protocol based on neurorehabilitation, including functional electrical stimulation (FES), in a clinical rehabilitation center. The second aim was to assess various parameters of FES according to the presence or absence of deep pain perception (DPP). Finally, another aim was to evaluate recovery under the intensive neurorehabilitation protocol (INRP) over time and define the limitations of this protocol. 

## 2. Material and Methods

This prospective observational cohort pilot study in cats with PTBPI was performed at the Arrábida Animal Rehabilitation Center (CRAA, Portugal) between 2016 and 2023 after approval by the Lusófona Veterinary Medicine Faculty Ethics committee, with signed consent from the cats’ owners.

### 2.1. Participants

All of the test subjects were domestic cats (*n* = 22) with previous traumatic injury (only from road traffic accidents) with acute non-progressive presentation that had been diagnosed with PTBPI. All had to present the following inclusion criteria: monoplegia with inability to support weight and fifth-degree lameness; shoulder extension and inability to extend the carpus and digits; presence of absence of ipsilateral hemiplegia (Figure 1); absent peripheral reflexes; and presence of absence of the cutaneous trunci reflex. 

In addition, subjects were required to have an absence of superficial pain distal to the elbow and lateral surface of the thoracic limb, while the medial side of the paw could have either absent or decreased pain perception. Deep pain could be decreased, present or absent. Lastly, all subjects had to show decreased triceps brachialis muscle tonus and could also present with Horner’s syndrome.

Additionally, to be selected, all cats had to have computed tomography or resonance imaging exams through which the brachial plexus injury was localized to a caudal lesion (C8, T1, T2). No conduction studies, EMG, functional tests (i.e., force plates) or nerve biopsies were conducted to evaluate the progress of the lesion. 

Exclusion criteria encompassed all other diseases that can affect the brachial plexus, such as inflammatory, infectious or neoplastic disease and other trauma lesions with different clinical presentations.

### 2.2. Study Design

All 22 cats with PTBPI were referred to the rehabilitation center three to seven days after trauma and were enrolled in this prospective observational cohort study following hemodynamic stabilization.

All had received a previous emergency-setting consultation in different hospitals, where they underwent stabilization involving restrictive resuscitation fluid therapy, blood work and point-of-care ultrasound. Upon admission to the rehabilitation center, all cats presented with a normal mental state, heart rate, respiratory rate, blood pressure and temperature. The neurorehabilitation examination involved the assessment of the following: passive standing posture; spinal reflexes of the thoracic limb (withdrawal reflex and extensor carpi radialis reflex); and palpation of all muscles that are innervated by the caudal brachial plexus nerves (Table 1), with specific palpation of the medial region of the thoracic limb, as described in Figure 2.

Key points of the neurorehabilitation exam used to assess C8-T2 injury are described in Table 2 [1,20].

The study design necessitated a specific examination of the brachial plexus region, given that EMG was not performed as an outcome measure. Therefore, it was critical to assess superficial pain and map the thoracic limb dermatomes to evaluate the presence of deep pain in the first and fifth digits and to monitor for paresthesia by tracking behavior changes, such as biting, licking or self-mutilation. The evaluation of superficial and deep pain was performed with a 12 cm Halsted mosquito forceps, and an ink marker was used to draw the cut-off point of the dermatomes map for further evaluation (Figure 3).

For these reasons, neurorehabilitation examinations were repeated each week to assess neurological changes. Also, possible iatrogenic outcomes of the protocol, such as burns, pruritus, hyperemia and signs of worsening paresthesia, were evaluated and recorded. 

### 2.3. Procedures

The neurorehabilitation protocol for PTBPI had multiple aims: (a) to maintain/restore the joint’s range of motion and neuromuscular function, preventing pain, trauma and self-mutilation [41]; (b) to prevent muscle contractures [39,42] and joint contractures, maintaining the elbow and carpus flexion/extension end-feel [43]; (c) to minimize neurogenic atrophy, promoting muscle strengthening and muscle flexibility; (d) to reduce the possibility of paresthesia and promote recovery of sensation [44]; (e) to achieve ambulation and improve quality of life. 

All cats (*n* = 22) were subjected to the same early INRP. Procedures took place six times per day in a controlled and quiet environment in the cat´s rehabilitation room, with different modalities, such as laser therapy and ultrasounds, followed by a 30 min window in which to perform the electrotherapy protocols (functional electrostimulation of the radial nerve) in association with locomotor training, according to each patient´s cardiorespiratory condition and phase of neurological recovery. 

All animals were hospitalized and were clinically discharged when they achieved ambulation, although the maximum time given to this study´s protocol was two months. Furthermore, all cats had their nails cut and wore Elizabethan collars to prevent self-mutilation, and wore harnesses during the administration of the protocol. Positive reinforcement was given in the form of treats, toys or play with other friendly cats. Modalities and exercises were executed by three veterinarians, a certified canine rehabilitation practitioner (CCRP) and two CCRP students. All evaluations were performed by a veterinarian diplomate of the European College of Veterinary Sports Medicine and Rehabilitation with a PhD in neurorehabilitation.

#### 2.3.1. Intensive Neurorehabilitation Protocol (INRP) 

##### 1st–3rd Week of INRP

In the first three weeks the protocol was similar (Table 3). It was essential to avoid active movements with limb stretching, and the main focus was to increase depolarization of the radial, median and ulnar nerves. This aim was achieved using class IV laser therapy (LiteCure Companion Therapy Laser^®^, New Castle, DE, USA), which was selected for its regenerative properties, applied to the anatomic pathway of the nerves. In addition, a 4-point technique using the same laser was applied to the carpal, elbow, and shoulder joints, promoting analgesia, anti-inflammatory effects, and edema reduction [45]. 

Additionally, the ultrasound modality (BTL-4820 Smart^®^, Hertfordshire, UK) was prescribed to promote neural depolarization [46], beyond its primary role in muscle relaxation and ligament flexibility. This treatment aimed to reduce secondary muscle contractures [47] and allow the use of passive range-of-motion exercises for the shoulder, elbow and carpal joints (Figure 4). These exercises could be repeated 4–6 times/day (10–30 sets per session), finishing with manipulation of all digits, individually or jointly [48].

FES (BTL-4820 Smart^®^, Hertfordshire, UK) was implemented based on the anatomic pathway of the radial nerve, with one electrode placed to stimulate the C6, C7, C8, T1 and T2 nerve roots and another placed in the motor point of the triceps brachialis muscle (Figure 5). Electrical parameters were selected according to DPP guidelines (Table 3). 

All rehabilitation modalities and exercises were immediately followed by the application of a corrective carpal splint (Buster Leg Splint, Kruuse, Denmark) with vet-wrap tape (Vet-Flex, Kruuse, Denmark), which was placed for 2 h in the morning and afternoon (Figure 6) to prevent joint contracture.

**Table 3 animals-14-00323-t003:** Intensive neurorehabilitation protocol in the 1st, 2nd and 3rd weeks.

Rehabilitation Modality/Exercise	Parameters	Implementation
Laser therapy (A)	18–22 J/cm^2^ Class IV laser Radial nerve pathway; SID	Regenerative
	5–10 J/cm^2^ Class IV laser 4-point joint technique; SID (Shoulder, Elbow, Carpus)	Analgesia, Anti-inflammatory effects
FES of the radial nerve (B)	30–40 Hz; 6–16 mA; 200 µs [49] Trapezoidal modulation 1:4 duty cycle 2–4 s ramp up; 8 s plateau; 1–2 s ramp down; 10 min; TID	Deep pain
	30–40 Hz; 6–24 mA; 200 µs [49] Trapezoidal modulation 1:4 or 1:5 duty cycle 2–4 s ramp up; 8 s plateau; 1–2 s ramp down; 10 min; TID	No deep pain
Range-of-motion exercises (C)	10–30 sets 4–6 times/day	All joints: shoulder, elbow, carpal and digits
Postural standing position (D)	2–3 min 4–6 times/day	
Ultrasound (E)	1 MHz; 1.5 w/cm^2^; 10 min Pulsed mode; Duty cycle of 20%; Pulse ratio 1:4; 5 cm transducer head	Muscles: triceps brachialis; biceps brachialis; extensor carpi radialis; lateral and common digital extensor

FES: functional electrical stimulation; J: Joules; Hz: hertz; mA: milliamperes; s: seconds; MHz: megahertz.

Regarding frequency, the sequence B + C+ D (Table 3) was repeated three times daily, whereas the other modalities were performed only once a day. 

ES on the pelvic limb was also performed to promote distal depolarization if the cat showed signs of proprioceptive deficits in the ipsilateral pelvic limb. This therapy included the sciatic nerve FES, with one electrode placed to stimulate the L7-S1 nerves and the other on the motor point of the flexor muscle group (50 Hz; 6–24 mA for cats without deep pain; 6–14 mA for cats with deep pain), as well as stimulation of the peroneal nerve branch (one electrode placed to stimulate nerves L7-S1 and the other on the dorsal region of the paw, with the same parameters) (Figure 7).

Between the second and third week, if a cat recovered joint motion of the elbow, stimulation of the distal radial nerve was performed, aiming to improve mobility of the carpal joint and reduce the possibility of contracture. Thus, radial nerve FES was performed for carpal extension, with one electrode at C7, C8, T1 and T2 and the other one on the dorsal region of the paw (Figure 8). 

##### 4th–6th Week of INRP

The laser therapy program continued, along with the radial nerve FES (both triceps muscle and extension of the carpus), range-of-motion exercises, postural standing, ultrasounds and, in some cases, FES of the pelvic limb. All these therapies were followed by locomotor training on the land treadmill (10–20 min), with bicycle movements performed by a CCRP veterinarian or nurse. This training was repeated twice a day, six days/week. Bicycle movements were performed without limb stretching and with direct contact between the digits and the treadmill belt to stimulate mechanoreceptors and proprioceptors (Figure 9). There was no incline, and the speed was increased until it reached 1.8 km/h. 

Around the 4th week, when the possibility of neural regeneration increased, underwater treadmill (UWTM) training was introduced and undertaken once a day for a maximum of 10 min, with the water level at the lateral epicondyle of the femur, at a temperature of 24–26 °C, with no incline and a maximum speed of 2.2 km/h (Figure 10). UWTM training was performed after the rehabilitation modalities of laser, ultrasound and FES, allowing physiological muscle contraction based on the agonist/antagonist rule. 

In this exercise, as in all types of exercises, the cats had to be in a controlled, calm and quiet environment and were motivated by positive reinforcements. The rehabilitator was inside the UWTM, performing rhythmic bicycle movements. 

Furthermore, in this neurological phase, postural standing exercises on top of a balance board were added (2 min, 4 min rest), followed by walking on different floor surfaces (2 min, 6 min rest) and stepping over cavaletti rails (1–2 min), always with the corrective splint applied in order to facilitate elbow flexion. 

If cats developed wounds, cleaning was performed at the end of the day with a 3% chlorhexidine solution and class IV laser therapy was done to stimulate granulation tissue [50].

##### 6th–8th Week of INRP

The same protocol was carried out, including components A + B + C + D + E (Table 3), with the addition of locomotor training on the land treadmill with a 10% incline and the application of a TheraBand to increase muscle strength and reduce neurogenic atrophy. The aim was to carry out these therapies for 30 min, twice a day; however, for patients that could not sustain this prolonged training, a comprehensive circuit was substituted, with the following components: stimulation while standing on a central pad (2 min), rest (2 min), walking on different floor types (2 min), rest (2 min), walking up/down stairs (2 min), rest (2 min), walking up/down ramps (2 min), rest (2 min), stepping over cavaletti poles (2 min), rest (2 min), walking in figures of eight, rest (2 min) and, lastly, free time (playing with other cats or with toys). 

In the last two weeks, the frequency of application of modalities such as laser and ultrasound were reduced to every 48 h and the laser was applied only to the interphalangeal and carpal regions of the joints. Additionally, the frequency of FES protocols was reduced to once a day, 5–6 times/week.

##### Neuropathic Pain

In addition, cats with signs of neuropathic pain were given pharmacological support in the form of pregabalin (2 mg/kg BID) for four weeks, which was then decreased to SID for the next four weeks. If any of these cats showed signs of pain, such as excessive licking, interferential transcutaneous electrical nerve stimulation (TENS) was prescribed for the carpal and digits regions. This technique used two different channels crossing each other at a 90◦ angle: firstly, two channels at 80–150 Hz, 0.5–1 mA, pulse duration 50 µs for 10 min; secondly, two channels at 1–10 Hz, 0.5–1 mA, pulse duration 150 µs for 10 min [51].

### 2.4. Data Collection

Data collected from the 22 cats included all variables mentioned in Table 4. 

At the follow-up time points the muscle mass (improved or not) and muscle weakness (improved or not) were also evaluated. The cats were considered ambulatory when they could rise and take at least 10 consecutive weight-bearing steps unassisted and without falling [52,53].

### 2.5. Statistical Analyses 

Data were recorded using Microsoft Office Excel 365^®^ (Microsoft Corporation, Redmond, WA, USA) and processed in IBM SPSS Statistics 25^®^ (International Business Machines Corporation, Armonk, NY, USA) software. A Shapiro–Wilk normality test (for n < 50), arithmetic means, median, mode, variance, standard deviation (SD), minimum, maximum and standard error of mean (SEM), were calculated and recorded for the continuous variables age and weight. Descriptive statistics with frequency analysis were calculated for all categorical nominal variables. Chi-square tests were also performed to verify relevant relationships, which were considered significant at a *p*-value < 0.05.

## 3. Results

Out of 22 individuals observed in this study, 68.2% were males and 31.8% were females. The mean age and weight of the individuals treated were 4.86 and 4.73 kg, respectively (Table 5).

Only 86% of the cats tested positive for dermatomes; the remaining 14% tested negative (Table 5). Nevertheless, all the cats had dermatomes between the elbow and carpus and between the carpus and digits.

Regarding the presence of dermatomes at T0, there was no significant relationship with ambulation at discharge; however, the recovery of dermatomes during rehabilitation was significantly associated with ambulation recovery [X^2^ (1, n = 22) = 9.263, *p* = 0.002]. Of the 72.7% of cats that recovered ambulation, all had positive dermatomes at medical discharge. These same 16 cats recovered movement of all three joints during the INRP: 11 within 30 days, 3 cats within 30–45 days and only 2 within 45–60 days. 

The evaluation of deep pain at T0 revealed the absence of DPP in the first four digits in 45.5% of the cats, while the remaining 54.5% had doubtful DPP. The relationship between DPP at admission and ambulation achievement was highly significant [X^2^ (1, n = 22) = 9.900, *p* = 0.002], with all cats with doubtful DPP achieving ambulation.

Deep pain in the fifth digit was present in 40.9% and doubtful in 59.1% and showed no significant relationship with ambulation recovery. However, a significant relationship [X^2^ (1, n = 22) = 4.701, *p* = 0.030] was observed regarding development of carpal contractures, with most contractures seen on cats with doubtful DPP in the 5° digit (n = 9). 

Additionally, a strong relationship between the development of carpal contractures and time until medical discharge [X^2^ (1, n = 22) = 22.000, *p* < 0.001] was found, with all cats that presented this clinical sign being discharged at 60 days.

Among all treated cats, 72.7% achieved ambulation, whereas 27.3% did not recover. For these 16 cats, the median time to ambulation was 30 days, with 11 cats being medically discharged at day 30 and 5 cats discharged at day 60. 

In total, 50% of the cats were medically discharged at day 30, and all were ambulatory, recovering DPP by 15 days (n = 9) or between 15 and 30 days (n = 2). All showed the knuckling position during the INRP, which was followed by total recovery of reflexes, dermatomes and joint movement, with absence of carpal contracture, although there was diminished extension of the carpal joint. 

Of the remaining 50% that were medically discharged by day 60, only 5 cats recovered total DPP and developed a knuckling position followed by ambulation, although all presented positive changes in their reflexes and the development of carpal contractures. There were six cats with incomplete recovery of DPP but with improvement in the fifth digit, and three of them recovered all dermatomes with joint movement until the elbow. These cats were treated with surgical arthrodesis. The other three recovered dermatomes only between the shoulder and the elbow and movement through the elbow, with amputation of the limb on later follow up (2 cats in F3 and 1 cat in F4).

Regarding neuropathic pain, 18.2% cats showed this clinical sign at admission. Later during the INRP, 50% cats showed development of neuropathic pain by day 15, which was associated with the development of carpal contractures [X^2^ (1, n = 22) = 18.333, *p* < 0.001]. These cats showed progressive improvement, and only six cats still showed this sign at medical discharge, these being the same cats that later required surgical arthrodesis or amputation.

The clinical problems that arose during the study period were wounds on the dorsal region of the digits that occurred due to neurological deficits and were recorded in 40.9% of the cats. No iatrogenic clinical problems were observed as a consequence of the INRP, including burns, pruritus or hyperemia. 

Horner syndrome was diagnosed in 13.6 % of the cats, with total recovery in two cats by day 30, one that achieved ambulation and one needing arthrodesis. The cats that had not recovered from the Horner syndrome by day 60 ultimately required limb amputation at F3. 

On follow up, there was an evident improvement in both muscle mass and muscle weakness (Figure 11). However, three cats showed only a slight improvement on the first follow up (3 months), with continued neuropathic pain. These cats later required limb amputation.

Compared to the contralateral limb, the affected limbs always had less muscle mass and more muscle weakness at all follow up visits until one year (F3). However, after two years (F4), seven of the cats showed no difference between the limbs, an improvement that was maintained until four years after treatment (F6) with resolution of neuropathic pain. 

## 4. Discussion

In this study, 22 cats were referred to the neurorehabilitation center after PTBPI diagnosis due to road traffic accidents, which are common in both human patients and small animals [5,14,15,16,54]. All had acute non-progressive injury of the peripheral nervous system, and the aim was to begin nerve repair early to improve functional outcomes [10,55] and avoid muscle fibrosis and neurogenic atrophy. Our INRP implementation timeline was similar to those used in human medicine and began earlier than the median time of 14–13 days after injury that is described by Menchetti et al. (2020) [54]. Furthermore, the same study revealed a median duration of 60 days until recovery, with owners reporting improvement in five cats. By comparison, in our study, these 22 cats had a high probability of axonotmesis (equivalent to Sunderland grade II–IV) and the median duration until ambulation was half that seen in the previous study (30 days).

Protocols in human medicine are usually based on the beneficial role of locomotor training, which is one of the best approaches to treat peripheral nerve injury when administered in conjunction with other physical exercises and modalities, such as ES. The volume and intensity of training, with standardized and correctly selected frequencies of ES, can improve functionally and can be translated to veterinary medicine in the spirit of the “one health” concept [56,57,58,59].

Although differences can be seen in recovery according to age due to the decrease in depolarization and concomitant problems associated with older cats, there was no significant relationship between the final functional outcome and age in this study. This result can probably be explained by the fact that our sample of younger cats were similar in age, with a mean age of 4.86 (±0.467) years. The same was observed with weight, with a mean of 4.73 kg (SEM 0.239), and with sex and breed. 

Many factors can influence functional recovery following nerve injury. Beyond the time elapsed between trauma and reparative treatment, there is the distance from the injury site to the target muscle and the regeneration rate, which is usually around 1 mm per day. The combination of a long distance and slow regeneration can result in atrophy of both the distal nerve and the effector muscles, which may make functional recovery impossible. Also, on admission (T0), 45.5% of cats were without DPP in the first four digits and 54.5% had doubtful DPP, an important fact given that the literature points to the absence of DPP as the single neurologic sign most strongly associated with the need for amputation of the affected limb [54,60,61]. 

The present study was performed in a rehabilitation clinical setting and was strictly guided by the neurorehabilitation treatment of these cats, with the aim of achieving functionality of the limb and ambulation (performing 10 consecutive steps without muscle weakness and falling). On admission, all of the cats had absent or doubtful DPP, making nerve biopsies a risk factor for neurological deterioration, that could compromise future functionality. This risk was the main reason that owners chose not to authorize the procedure. 

Total ambulation recovery was 72.7% after the implementation of INRP, with a median time of 30 days: 11 cats discharged at day 30 and 5 cats at day 60. Thus, time to recovery was shorter than that previously described [52], although one cat recovered at the rate described by Santifort (2016) [62]. However, we cannot completely exclude the possibility that this fast recovery rate was due to spontaneous regeneration. 

Sensorimotor evaluation with dermatomes recovery showed a clear, significant association with ambulation throughout the study (*p* = 0.002) and was related to resolution of joint contracturesm with 11 cats recovering shoulder, elbow, and carpus motion within 30 days. Therefore, these results illustrate that it is essential to avoid muscle and joint contractures, maintaining limb mobility with conservative treatment [62,63,64,65]. 

In our results, there was a significant relationship between DPP and ambulation (*p* = 0.002): all cats with doubtful DPP became ambulatory. This result may be related to the conservative nature of the INRP treatment, which ensured protection of the injury region, avoiding further damage to nerve structures with pain control and management of sensory deficits in the first 15 to 21 days. The ES programs can be used as an adjunctive tool with early implementation, with stimulation administered according to the neurological evaluation and range of motion of the joint in the affected limb. 

Early recovery of DPP in the fifth digit showed a significant relationship with ambulation (*p* = 0.03), and 9 cats that developed carpus contractures due to faster re-innervation of the flexor muscles than of the extensor muscles exhibited a persistent knuckling position while walking [66,67]. Experimental studies on rats with peripheral lesions on the sciatic nerve suggest the implementation of passive range-of-motion and assisted active exercises, such as the use of a 45° net inside the cage to avoid prolonged muscle inactivation associated with incorrect positioning [68,69,70]. 

In human medicine, several non-surgical approaches, including pharmacological, electrical, cell-based and laser therapies have been used to promote re-myelination and enhance recovery in diseases of the peripheral nervous system [17,71,72,73,74]. The implementation of ES, particularly between the second and third week, may help muscles achieve passive contraction, promote normal electromyographic waves during treatment [75], reduce muscle atrophy and promote muscle reinnervation by increasing expression of structural protective proteins and neurotrophic factors [74]. 

Various authors [76,77,78] have suggested that it is possible to mimic the physiological wave of Ca^2+^ influx that generates a retrograde signal leading to subsequent activation of autonomous cellular mechanisms and thus initiating regeneration [79,80]. FES may in this way help to promote upregulation of BDNF, neurotrophins and TrK receptors [81,82], as well as glial-cell-like derived neurotrophic factor (GDNF) [74,83,84,85]. Additionally, studies have demonstrated earlier outgrowth of axons and Schwann cells, with accelerated reconnection to the target injury site. This result may be related to cellular mechanisms that increase the production of adenosine monophosphate with Ca2+ influx, regulating BDNF and TrkB expression [26,79,86,87,88].

In the present study, 50% of the cats recovered ambulation by day 30, but all showed progressive sensorimotor improvement. The cats required different ES programs, adapted to each case and their neurological evolution (i.e., presence of knuckling position). For example, FES for the extensor muscles of the carpus with a precise trapezoid current and parameters adjusted to DPP was used to generate an action potential strong enough for nerve depolarization and consequent muscle contraction [89]. Also, accurate placement of electrodes is crucial for opposing the low current that makes physiological contraction impossible under these conditions [90]. Cats with evident neurogenic atrophy need more time to recover; to avoid muscle fatigue in these cats, the programmed duty cycle should be 1:5, with a ramp-down time of 1–2 sec [91,92]. The trapezoid modulation, which is a triangular current with a duration of 200 μs, has shown better effects on denervated muscles than have other modulations [74,93].

Thus, in this study, FES was conducted with specific parameters and methods according to the daily neurological evaluation of each cat and consideration of the fact that high frequencies of ES can aggravate nerve damage [74,94], causing fatigue and neuropathic pain. 

The authors cannot state that the obtained outcomes are a direct result of the implementation of FES; however, a multimodal approach with ES and locomotor training may have had a fundamental role in their recovery. It would be interesting to assess recovery, specifically, the extension movement of the carpus, with functional measures (i.e., kinematic and kinetic analysis), which would allow evaluation of progress in muscle strength. However, as this was a clinical study, limb functionality was evaluated using neurological examination, the dermatomes map and behavioral changes indicating possible paresthesias. 

The INRP, which included ES and treadmill training, aimed to achieve synergetic BDNF upregulation [29,95,96,97,98]. Furthermore, the intensive locomotor exercise started after 21 days, when neural regeneration was higher, preventing muscle neurogenic atrophy and promoting improvement in strength [75,76]. After six weeks, the start of stretching maneuvers allowed to the cats to maintain muscular flexibility [75,99], and passive stretching promoted stimulation of mechanoreceptors, slowing protein degradation [100] and increasing the range of motion of the elbow and carpus. Among the other 50% of cats medically discharged at day 60, five recovered DPP and ambulation of the injured limb. However, six showed improved DPP only on the fifth digit, and three had dermatomes recovery with complete movement of the elbow, a perfect scenario for performing an arthrodesis. This treated improves functionality and quality of life and decreases neuropathic pain. 

Some authors suggest that recovery after PTBPI requires more than three months (from 3 to 12 months) [17,54]. Our results showed that only three of the cats did not recover dermatomes in the forearm with joint movement to the elbow. These cats showed ongoing neuropathic pain and presented with wounds and discomfort throughout the long-term follow-ups. Ultimately, 13.6% of the cats required limb amputation, in agreement with [54,60]. 

In human patients with brachial plexus injury, neuropathic pain is described in 30 to 80% [101,102] and has a highly refractory presentation [103,104,105,106,107]. In our sample population, at admission, 18.2% of the cats had this clinical sign, increasing to 50% during the first 15 days. This pain was probably associated with carpal contracture (*p* < 0.001), and in 40.9%, it was related to wounds on the dorsal region of the digits that arose due to their knuckling position. Pain management involved pregabalin administration, and resolution of contracture and knuckling improved the outcomes.

In this investigation, there was high variability in sensory patterns and sympathetic innervation consistent with the evolution of Horner syndrome, as well as weakness. In some cases, these symptoms followed a positive trajectory. During the long-term follow ups, seven cats showed improvements in muscle mass and muscle weakness at F4 (two years). These improvements were sustained at the four-year follow-up, and these cats did not develop neuropathic pain. 

Interferential TENS treatments can also be associated with progressive improvement in neuropathic pain, given their analgesic effects [75,108]; they can be used to control allodynia and hyperalgesia, according to the type of injury and the probability of recovery [75,109]. In addition, to treat pain, inflammation and edema, the INRP included class IV laser therapy, which acts via the upregulation of nitric oxide. Nitric oxide, like other free radicals that result from lipidic peroxidation of the central and peripheric nervous system, is associated with cell necrosis and has an especially important role on neuropathic pain compared to other human medicine peripheral neuropathies (e.g., diabetic polyneuropathy, multiple sclerosis, stroke) [101,110]. 

In our study, there were no observations of signs related to possible phantom limb pain, which is reported to occur in 54–85% of amputees in human medicine [98,111,112,113]. This phenomenon, interpreted as a result of cortical reorganization [109], is estimated to happen in veterinary patients within the first two years after amputation and can persist throughout life in up to 10% [114,115,116]. However, this phenomenon was not observed on long-term follow-ups up to four years for the three cats that underwent amputation. Additionally, no iatrogenic effects, including burns, pruritus or hyperemia, were observed during the INRP.

The main limitation of this study was the absence of EMG studies that can localize injuries and provide functional information and follow-up of the re-innervation process, although these studies require specific timing. Such timing is difficult to coordinate, mostly because, in axonal lesions, reduction of CMAPs requires several days, while electrical stimulation distal to the injury can be normal and signs of Wallerian degeneration take at least two to three weeks to appear. Subsequent research endeavors should prioritize the inclusion of EMG procedures as an integral component of INRP monitoring and early injury classification. As this was a clinical study, biopsies were considered an invasive procedure and were not allowed by the owners. However, this was a preliminary pilot study to verify the possibility of an intensive neurorehabilitation approach and was safe, repeatable and feasible. The intervention resulted in functional recovery, based on the neurorehabilitation examination. This study opens the possibility for future investigations with kinematic, kinetic and superficial electromyographic outcome measures. 

The small sample size is another limitation that resulted from the strict criteria for inclusion, which were needed to reduce selection bias. Additionally, there was no control group, as the study was carried out in a clinical rehabilitation center. Thus, it is important to mention that these findings should be interpreted with caution. 

## 5. Conclusions

Early INRP may have a role in promoting ambulation in cats diagnosed with partial traumatic brachial plexus injury and can be applied in a safe and repeatable way. 

Appropriate FES parameters, depending on DPP, can be essential to improving sensory-motor recovery by allowing expression of structural protective proteins and neurotrophic factors. The strength of these effects increases with the synergic effect of locomotor training, which in this study, started after 21 days. This pilot study achieved 72.7% ambulation, with a median time of 30 days, although one must always consider the possibility of spontaneous neural reorganization. This investigation should be continued with further studies that include specific outcome measures, such as electromyography. 

## Figures and Tables

**Figure 1 animals-14-00323-f001:**
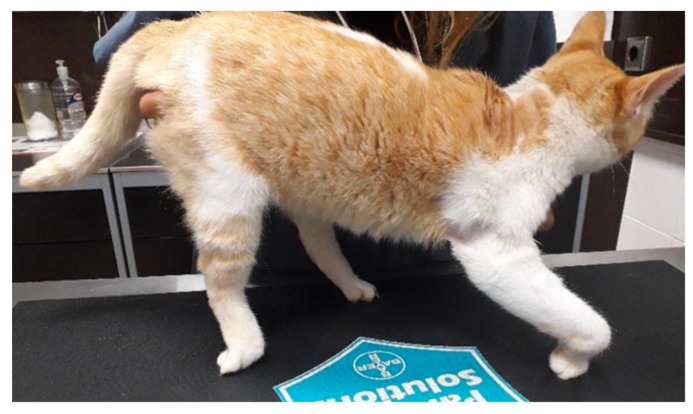
Cat with ipsilateral monoplegia and proprioceptive deficits, showing knuckling posture.

**Figure 2 animals-14-00323-f002:**
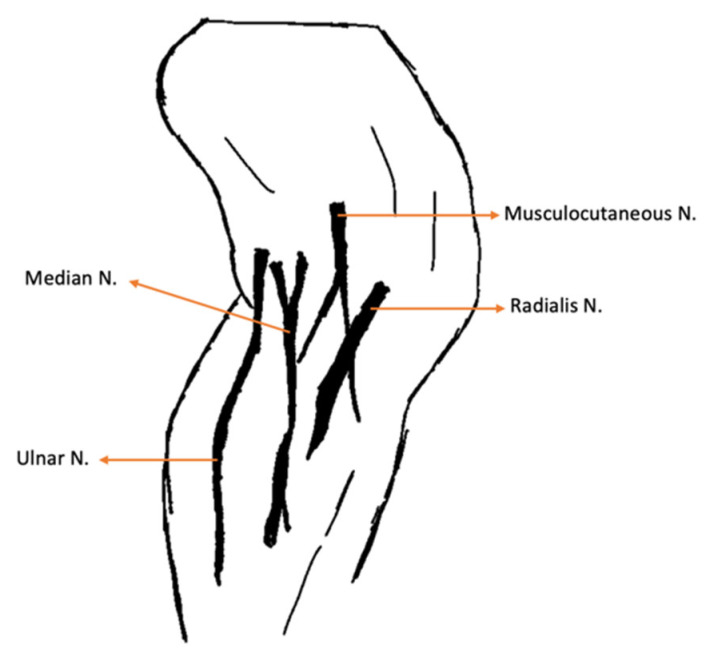
Medial aspect of the thoracic limb. N.: Nerve.

**Figure 3 animals-14-00323-f003:**
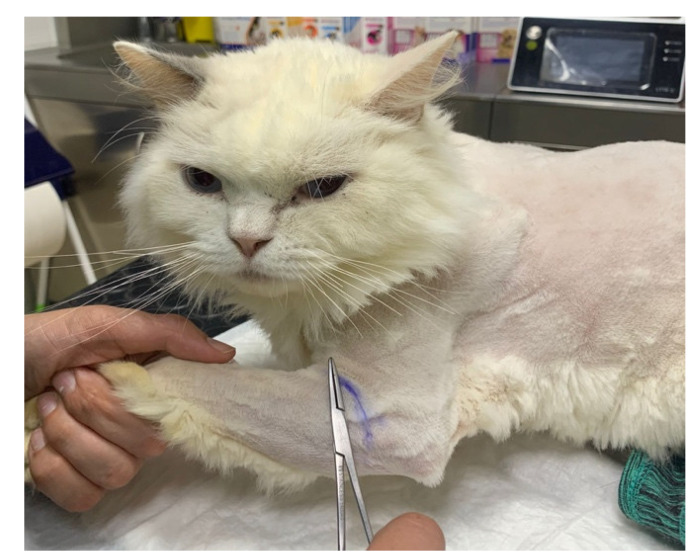
Cut-off point of the dermatomes map on the thoracic limb, marked with ink for further evaluation.

**Figure 4 animals-14-00323-f004:**
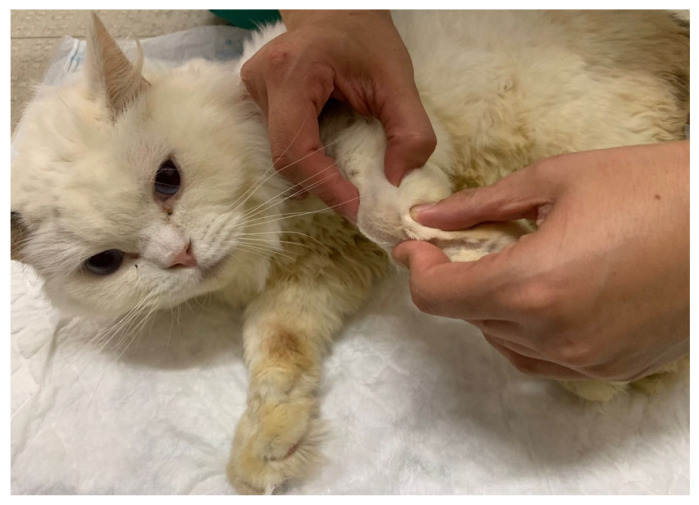
Passive range-of-motion exercises on the carpal joint of a cat.

**Figure 5 animals-14-00323-f005:**
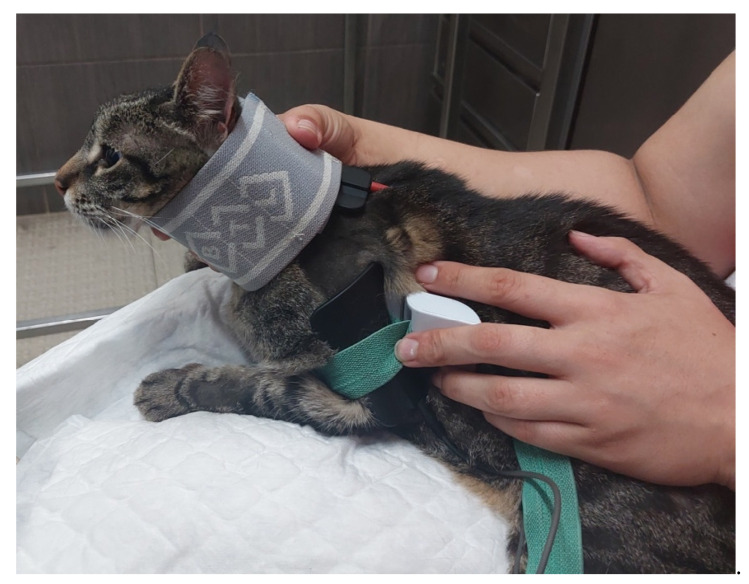
Functional electrical stimulation of the radial nerve (one electrode placed to stimulate the C6, C7, C8, T1 and T2 nerve roots and another placed in the motor point of the triceps brachialis muscle).

**Figure 6 animals-14-00323-f006:**
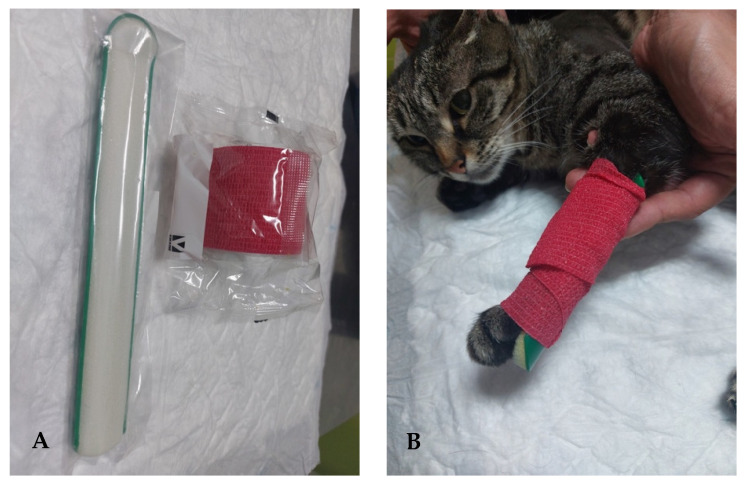
Corrective carpal splint. (**A**) Material for the splint. (**B**) Splint applied to the carpus of a cat.

**Figure 7 animals-14-00323-f007:**
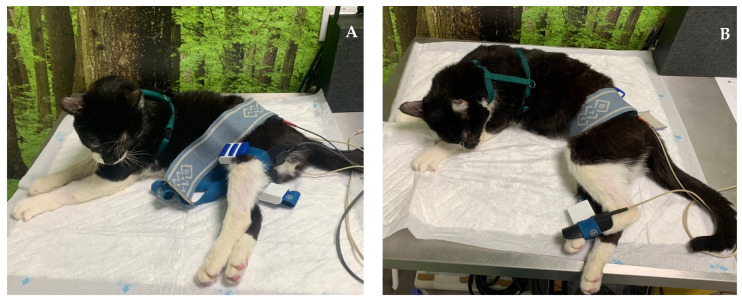
Functional electrical stimulation of the hindlimb. (**A**) Sciatic nerve stimulation (one electrode placed to stimulate L7-S1 and the other on the motor point of the flexor muscle group); (**B**) stimulation of the peroneal nerve branch (one electrode placed on L7-S1 and the other on the dorsal region of the paw).

**Figure 8 animals-14-00323-f008:**
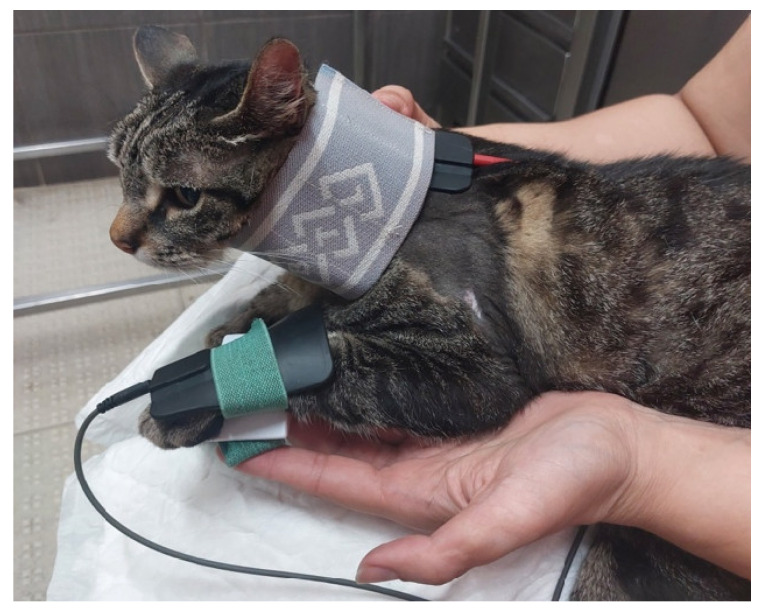
Functional electrical stimulation of the radial nerve for carpal extension (one electrode at C7, C8, T1 and T2 and the other one on the dorsal region of the paw).

**Figure 9 animals-14-00323-f009:**
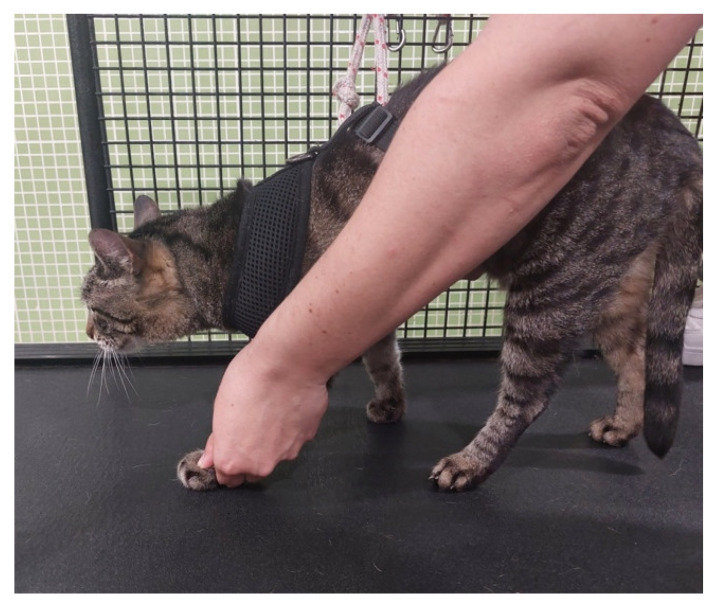
Locomotor training on a land treadmill, with bicycle movements performed on the affected limb.

**Figure 10 animals-14-00323-f010:**
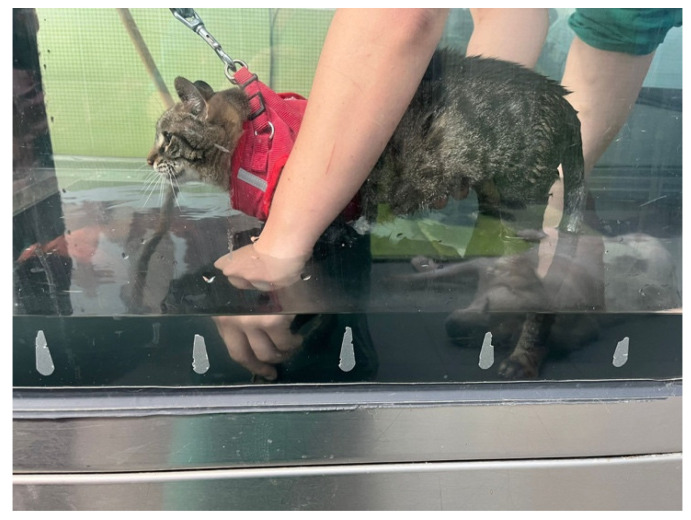
Locomotor training on the underwater treadmill, with bicycle movements performed on the affected limb.

**Figure 11 animals-14-00323-f011:**
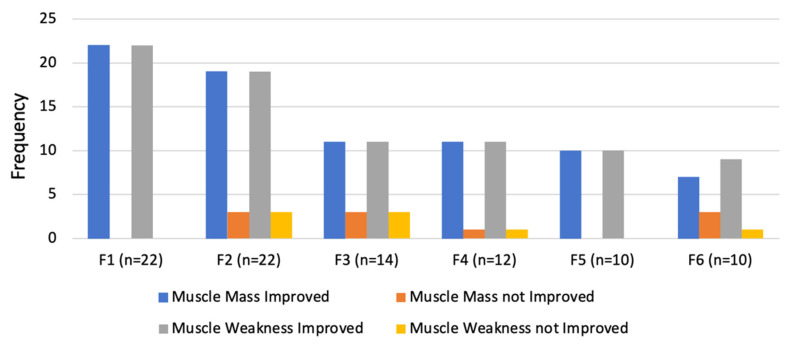
Evaluation of outcomes regarding muscle mass and muscle weakness in the follow-up consultations. F1: 3 months; F2: 6 months; F3:1 year; F4: 2 years; F5: 3 years; F6: 4 years.

**Table 1 animals-14-00323-t001:** Muscles innervated by the caudal brachial plexus nerves.

Nerve	Origin	Innervated Muscles
Radial	C6, C7, C8, T1, T2	Extensor carpi ulnaris; Triceps brachialis; Extensor carpi radialis; Lateral and common digital extensor
Ulnar	C8, T1, T2	Deep digital flexor; Flexor carpi ulnaris
Median	C7, C8, T1	Superficial digital flexor; flexor carpi radialis
Lateral thoracic	C8, T1	Cutaneous trunci

**Table 2 animals-14-00323-t002:** Key points of the brachial plexus neurorehabilitation examination [1,20].

Nerve	
Radial	Innervates the extensor muscle of the triceps group, which is responsible for the extension of the elbow and the muscles that extend both the carpus and the digits; These dermatomes are located in the dorsal and lateral antebrachium and dorsal paw.
Ulnar	The ulnar nerve carries sensory fibers to the skin on the caudal side of the thoracic limb and dorsolateral aspect of the fifth digit; Innervates the muscles that are responsible for carpus and digit flexion.
Median	Accompanied by the brachial artery and vein along most of its path; Innervates the muscles that are responsible for carpus and digit flexion.

**Table 4 animals-14-00323-t004:** Description and categorization of variables for statistical analysis.

Type of Variable	Variable Name	Description
Continuous quantitative	Age	<7 years or ≥7 years
	Weight	<5 kg or ≥5 kg
	Time until medical discharge	30 days or 60 days
	Time until DPP recovery	<30 days or ≥30 days
Categorical nominal	Sex	Male or female
	Breed	Purebred or Mixed breed
	DPP 1st–4th digits DPP fifth digit	Present, absent or doubtful
	Withdrawal reflex Extensor carpi radialis reflex Cutaneous trunci reflex	Present, absent or decreased
	Horner syndrome	Present or absent
	Knuckling	Present or absent
	Muscle atrophy (triceps and extensor carpi radialis)	Present, absent or mild
	Dermatomes (up to the elbow, between the elbow and carpus, between carpus and digits)	Present or absent
	Shoulder motion Elbow motion Carpus motion	Present or absent
	Carpal contracture	Present, absent or mild
	Standing position	Present or absent
	Arthrodesis	Performed or not performed
	Ambulation	Present or absent
	Neuropathic pain	Present or absent
	Clinical occurrences	Present or absent

DPP: Deep pain perception.

**Table 5 animals-14-00323-t005:** Characterization of study population (n = 22) at time of admission.

Variable	Category	Percentage of Individuals (n = 22)	Mean ± SE (SWNT)
Age	<7 years old ≥7 years old	72.7 27.3	4.86 ± 0.467 (0.021)
Weight	<5 kg ≥5 kg	45.5 54.5	4.73 ± 0.239 (0.071)
Sex	Male Female	68.2 31.8	-
Breed	Mixed	86.4	-
	Persian	13.6	
Knuckling	Forelimb	Absent: 100	-
	Hindlimb	Absent: 90.9 Present: 9.1	
DPP	1–4th Digits	Absent: 45.5 Doubtful: 54.5	-
	fifth Digit	Present: 40.9 Doubtful: 59.1	
	Withdrawal reflex	Absent: 100	
Reflexes	Extensor carpi radialis reflex	Absent: 100	-
	Cutaneous trunci reflex	Absent: 27.3 Present: 72.7	
Horner syndrome	Absent: 86.4 Present: 13.6		
			-
	Up to the elbow	Absent: 13.6 Present: 86.4	-
Dermatomes	Between elbow and carpus	Absent:100	
	Between carpus and digits	Absent: 100	
Joint motion	Shoulder	Absent: 13.6 Present: 86.4	-
	Elbow	Absent: 100	
	Carpus	Absent: 100	
Muscle atrophy	Triceps brachialis	Absent: 86.4 Present: 13.6	-
	Extensor carpi radialis	Absent: 100	
Carpal contracture	Absent: 100		-
Standing position	Absent: 100		-

DPP: Deep pain perception; SWNT: Shapiro–Wilk normality test; SE: standard error.

## Data Availability

The data presented in this study are available upon request from the corresponding author.

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
