# Peer review of "The Role of Early Rehabilitation and Functional Electrical Stimulation in Rehabilitation for Cats with Partial Traumatic Brachial Plexus Injury: A Pilot Study on Domestic Cats in Portugal"

_animals, 2024, doi:10.3390/ani14020323_

Round 1

Reviewer 1 Report (Previous Reviewer 2)

Comments and Suggestions for Authors

 Thank you for addressing the comments by this reviewer, the paper is much improved and is a good addition to the rehabilitation literature. Thank you for taking the recommendations seriously and for improving the readability and the discussion within the paper.

Comments on the Quality of English Language

There are still a few small issues that may be able to be handled by the individuals that typeset the article. Commas are used instead of periods for decimal points which is common in some countries but not in others and I think that may be confusing to some. Overall the English is much better. 

Author Response

Reviewer 1

Thank you for addressing the comments by this reviewer, the paper is much improved and is a good addition to the rehabilitation literature. Thank you for taking the recommendations seriously and for improving the readability and the discussion within the paper.

“There are still a few small issues that may be able to be handled by the individuals that typeset the article. Commas are used instead of periods for decimal points which is common in some countries but not in others and I think that may be confusing to some. Overall the English is much better.”

We kindly appreciate the reviewer´s comments and taking in consideration the suggestion, we change all the commas for periods in the decimal points.

Reviewer 2 Report (New Reviewer)

Comments and Suggestions for Authors

Comments and Suggestions to the Authors

The pilot study on ‘The role of early rehabilitation and functional electrical stimulation in partial traumatic brachial plexus injury cats’ diagnosed the partial traumatic brachial plexus injury (PTBPI) among 22 cats, aiming to explore how far the response of an early intensive neurorehabilitation protocol in a clinical setting is useful in ambulation. The results on treatment show that a majority (73%) of the cats achieved ambulation, with 9 cats within 15 days, 2 cats within 30 days and 5 cats within 60 days. Therefore, it is a new and very useful findings, management of domestic cats. However, the implications of this study also have relevance for wild cats, which are also increasingly affected by vehicular traffic, when they cross the highways that cut across many Protected Areas across the Globe, though there are limitations in handling the free-ranging cats.

Although there is no major issue in the manuscript, there are some ambiguities that need clarity, there are unnecessary statements in the text and duplications in the table that can be eliminated. And these are given below, as per line no and incorporating them will not only improve the clarity by will also compress and comprehend the MS.

Line No. 2-3: The role of early rehabilitation and functional electrical stimulation in partial traumatic brachial plexus injury cats: a pilot study

In the above title instead of injury cat, injured cat would be more appropriate.

Secondly, giving some clue about whether this study is on wild or domestic cats and also the broad region study area at the title itself, also gives the reader the intensity of such issues faced by free-ranging or domestic or by both cats. The details could be added in the second part of the title for example, A pilot study on domestic cats of Portugal.      

Line No: 35: 22 cats diagnosed with partial traumatic brachial plexus injury (PTBPI),

In the above 22 out of ?? cats diagnosed will also show how common the issue concerned is? about which the MS talks about in line nos. 31 and 55.

Line No. 41 Results have shown…….until 60 days.

Instead of the above, Results on treatment have shown….. until 60, would add importance to the treatment itself. Further, instead of ‘until’ it is better to use ‘within’.   

Line No. 153-154: All were from previous traumatic injury (only from road traffic accidents) with acute non-progressive presentation.

Please add, details about whether these 22 cats are domestic or also free-ranging individuals too. Because the free-ranging cats too face a similar situation. Even if the present study applicable to domestic cats, the implications of the study applicable to free-ranging species too.

Line No. 486-504: All these details of continuous and categorical variables could be placed in a table with three columns (column one for type of variable: continuous and categorical, column two showing the details each variable name and column three showing the description each variable and if possible, wherever applicable, how each variable was measured in a separate column), so that it is easier to refer and understand about each variable than going through the long sentence of text.

Line No. 515-518: Much of the information described are pertain to the method part, which are given in method section. So, describe your observed results in a straight forward manner, without the method component.

Out of 22 individuals observed in this prospective cohort pilot study, 68.2% were males and 31.8% were females. The mean age and weight of the individual treated were 4.86 and 4.73 kg respectively (Table 4).

 In the last column of Table 4, most of the figures mentioned are with a coma in between instead of a full stop. For example, the mean age shown as 4,86 should be 4.86. The same way, check entre each  figure.  Further, the SD and SEM are basically deals with similar details, therefore better to restrict with anyone, ideally SE.  Same way, I do not think it is necessary to incorporate all the details from a descriptive statistic, unless it essential and are going to deal about. For example, the mean, median and mode. Normal procedure is to show mean ± SE (n), all these can be placed in a single column and add the details of these variable category (i.e. < 7 years old: 72.7%;  ≥ 7 years old: 27.3%) and some of the details, what you have placed in Table 5 into the Table 4 itself adding a column called ‘variable category’. See for example the table shown below. 

Variable

Category

Percentage of individuals (n = 22)

Mean± SE (SWNT*)

Sex

Male

68.2

???

Female

31.8

Age

< 7 years old

72.7

4.86 ± 0.467 (0.021)

> 7 years old

27.3

Breed

Mixed

86.4

???

Persian

13.6

Weight

<5 kg.

> 5 kg.

* Shapiro-Wilk Normality Test value

In a table all columns need to be labelled. You cannot label one and ignore other columns.  For example, there is no column label for the first column both in Table 4 and 5. Both % value and frequency value of the % are duplications. When you give % for a given frequency, you should give the total frequency that is the ‘n’ along with column label, as shown in column 3 of above table, so that % percentage in each category (column 2) is out of total n, which is 22 in this study could be made out by the readers.

A table should be compact and informative and thus the MS will appear very compact. A MS with too many or too long tables, unless it is essential, too many pages, won’t be eye-catching and thus the MS will be ignored by the readers. Further, the space is expensive for the journal, even for an online version, and thus space needed to be saved.

Line No. 521-524 seem to be included into the MS only for citing Table, which need to be deleted and cite the Table 5 in Line No. 528-529 and modify or condense the line as follows.

Only 86% of the cats were tested positive and the rest 14% showed negative for dermatomes (Table 5). Nevertheless, all the cats had dermatomes between elbow and carpus and between carpus and digits.

Line No. 552: however the recovery of dermatomes during rehabilitation presented a strong significance [X2 (1, n=22) = 9.263, p=0.002]. What does mean by significance needs to be described instead simply saying significance, everywhere.

Please delete all frequency figures out of the total for example in line number 533: 16/22 shown after 72.7%. Either one is more than sufficient.

In Figure 11: The legend for Muscle Mass improve and Muscle Mass not improve to be changed into Muscle mass improved and Muscle mass not improved and Muscle Weakness also modified as per the suggestion made for Muscle mass.

I do not also any see axis label.

The bar diagram need not be in 3D, it can be simple bar.

Line No. 615: with a mean age of 4.86 years (SEM = 0.467). Here add the SEM with the mean including the ± in between. After stating about the mean and SE, justify why a significant relation between the final functional outcome and age is expected. Also say whether the mean sampled age indicates young or middle or older age-class, with which what functional response is expected and what you have observed.

 Line No. 762: This investigation should be continued with further studies.  Instead of saying continued further studies, say with what additional components or aspects? As you have mentioned the limitation, so that the measure suggested is comprehensive.

Author Response

Reviewer 2

The pilot study on ‘The role of early rehabilitation and functional electrical stimulation in partial traumatic brachial plexus injury cats’ diagnosed the partial traumatic brachial plexus injury (PTBPI) among 22 cats, aiming to explore how far the response of an early intensive neurorehabilitation protocol in a clinical setting is useful in ambulation. The results on treatment show that a majority (73%) of the cats achieved ambulation, with 9 cats within 15 days, 2 cats within 30 days and 5 cats within 60 days. Therefore, it is a new and very useful findings, management of domestic cats. However, the implications of this study also have relevance for wild cats, which are also increasingly affected by vehicular traffic, when they cross the highways that cut across many Protected Areas across the Globe, though there are limitations in handling the free-ranging cats.

Although there is no major issue in the manuscript, there are some ambiguities that need clarity, there are unnecessary statements in the text and duplications in the table that can be eliminated. And these are given below, as per line no and incorporating them will not only improve the clarity by will also compress and comprehend the MS.

Thank you so much for all the reviewer´s comments, we do think we could cope with all of these suggestions and that they will for sure improve our manuscript.

Line No. 2-3: The role of early rehabilitation and functional electrical stimulation in partial traumatic brachial plexus injury cats: a pilot study

In the above title instead of injury cat, injured cat would be more appropriate.

As suggested by the reviewer we change injury for injured cat.

Secondly, giving some clue about whether this study is on wild or domestic cats and also the broad region study area at the title itself, also gives the reader the intensity of such issues faced by free-ranging or domestic or by both cats. The details could be added in the second part of the title for example, A pilot study on domestic cats of Portugal.      

We kindly appreciate the reviewer´s comment and, as suggested, we added in the title “A pilot study on domestic cats of Portugal.”

Line No: 35: 22 cats diagnosed with partial traumatic brachial plexus injury (PTBPI),

In the above 22 out of ?? cats diagnosed will also show how common the issue concerned is? about which the MS talks about in line nos. 31 and 55.

We thank the reviewer for this comment and would like to explain that, although we have many neurological cats in our rehabilitation center and this is one of the common injuries, the only cats that were included in the study were these 22 cats. In this period, they were the only cats that were submitted to the protocol accepted by the owners.

Line No. 41 Results have shown…….until 60 days.

Instead of the above, Results on treatment have shown….. until 60, would add importance to the treatment itself. Further, instead of ‘until’ it is better to use ‘within’.   

 As suggested by the reviewer, we changed and have re-written the sentence.

Line No. 153-154: All were from previous traumatic injury (only from road traffic accidents) with acute non-progressive presentation.

Please add, details about whether these 22 cats are domestic or also free-ranging individuals too. Because the free-ranging cats too face a similar situation. Even if the present study applicable to domestic cats, the implications of the study applicable to free-ranging species too.

We kindly appreciate this comment and added the term “domestic cats” in the sentence. This is because all were domestic cats that have street access but there was not any free-ranging species.

Line No. 486-504: All these details of continuous and categorical variables could be placed in a table with three columns (column one for type of variable: continuous and categorical, column two showing the details each variable name and column three showing the description each variable and if possible, wherever applicable, how each variable was measured in a separate column), so that it is easier to refer and understand about each variable than going through the long sentence of text.

 As suggested by the reviewer, lines from 486 to 504 were re-written into a table.

Line No. 515-518: Much of the information described are pertain to the method part, which are given in method section. So, describe your observed results in a straight forward manner, without the method component.

Out of 22 individuals observed in this prospective cohort pilot study, 68.2% were males and 31.8% were females. The mean age and weight of the individual treated were 4.86 and 4.73 kg respectively (Table 4).

As suggested by the reviewer, we re-write this paragraph as suggested (line 518).

In the last column of Table 4, most of the figures mentioned are with a coma in between instead of a full stop. For example, the mean age shown as 4,86 should be 4.86. The same way, check entre each figure.  Further, the SD and SEM are basically deals with similar details, therefore better to restrict with anyone, ideally SE.  Same way, I do not think it is necessary to incorporate all the details from a descriptive statistic, unless it essential and are going to deal about. For example, the mean, median and mode. Normal procedure is to show mean ± SE (n), all these can be placed in a single column and add the details of these variable category (i.e. < 7 years old: 72.7%;  ≥ 7 years old: 27.3%) and some of the details, what you have placed in Table 5 into the Table 4 itself adding a column called ‘variable category’. See for example the table shown below.  

Variable

Category

Percentage of individuals (n = 22)

Mean± SE (SWNT*)

Sex

Male

68.2

???

Female

31.8

Age

< 7 years old

72.7

4.86 ± 0.467 (0.021)

> 7 years old

27.3

Breed

Mixed 

86.4

???

Persian

13.6

Weight

<5 kg.

> 5 kg.

* Shapiro-Wilk Normality Test value

In a table all columns need to be labelled. You cannot label one and ignore other columns.  For example, there is no column label for the first column both in Table 4 and 5. Both % value and frequency value of the % are duplications. When you give % for a given frequency, you should give the total frequency that is the ‘n’ along with column label, as shown in column 3 of above table, so that % percentage in each category (column 2) is out of total n, which is 22 in this study could be made out by the readers.

A table should be compact and informative and thus the MS will appear very compact. A MS with too many or too long tables, unless it is essential, too many pages, won’t be eye-catching and thus the MS will be ignored by the readers. Further, the space is expensive for the journal, even for an online version, and thus space needed to be saved.

We kindly thank the reviewer for all these comments regarding the tables and the results. As suggested, we changed table 4 and 5 considering all it was mentioned. Also, we appreciate all of the statistical tips that made this section clearer and easy to read.

Line No. 521-524 seem to be included into the MS only for citing Table, which need to be deleted and cite the Table 5 in Line No. 528-529 and modify or condense the line as follows.

Only 86% of the cats were tested positive and the rest 14% showed negative for dermatomes (Table 5). Nevertheless, all the cats had dermatomes between elbow and carpus and between carpus and digits.

As suggested by the reviewer, we re-write this paragraph (line 558).

Line No. 552: however, the recovery of dermatomes during rehabilitation presented a strong significance [X2 (1, n=22) = 9.263, p=0.002]. What does mean by significance needs to be described instead simply saying significance, everywhere.

Please delete all frequency figures out of the total for example in line number 533: 16/22 shown after 72.7%. Either one is more than sufficient.

We kindly appreciate this comment, and as suggested, we have explained what does significance mean in the MS and removed all frequencies.

In Figure 11: The legend for Muscle Mass improve and Muscle Mass not improve to be changed into Muscle mass improved and Muscle mass not improved and Muscle Weakness also modified as per the suggestion made for Muscle mass.

I do not also any see axis label.

The bar diagram need not be in 3D, it can be simple bar.

As suggested by the reviewer, we change the figure considering all it was mentioned.

Line No. 615: with a mean age of 4.86 years (SEM = 0.467). Here add the SEM with the mean including the ± in between. After stating about the mean and SE, justify why a significant relation between the final functional outcome and age is expected. Also say whether the mean sampled age indicates young or middle or older age-class, with which what functional response is expected and what you have observed.

Thank you for these comments, we have re-written the paragraph for better understanding.

For better explanation, it is expected to have a decreased in depolarization in older patients, decreasing recovery. However, this was not expected in our manuscript because our mean age was younger patients, a mean of 4.86.  Our main sample was younger cats because these are the ones that have access to the street and most probable to be involved in car accidents, although in our clinical experience older cats that are submitted to similar protocols also respond positively.

 Line No. 762: This investigation should be continued with further studies.  Instead of saying continued further studies, say with what additional components or aspects? As you have mentioned the limitation, so that the measure suggested is comprehensive.

As suggested by the reviewer we have added additional components that are important for further studies in line 799.

Reviewer 3 Report (New Reviewer)

Comments and Suggestions for Authors

Dear authors.

I have carefully reviewed the current manuscript. It is a really interesting study in a common disorder in cats, however the presentation of the manuscript as a whole, is bad.

In the current form, the manuscript is not suitable for publication. The Materials and Methods section should be re-written. Many information mentioned in the Materials section should be in the Results section. Many of the Results are abruptly mentioned.

I have made some recommendations below, however you have to make a substantial progress in revising the manuscript, in order to be appropriate for publication.

Line 139: There is no need to mention the type of the study there (prospective, observational, cohort pilot study). That belongs to the Materials and Methods section, as you have already done.

Line 151: in this section, you should mention the number of the cats participated in the study

Line 153: ''had'' instead of ''were from''

Line 153-167: The way you present the inclusion criteria is totally wrong and should be re-written. The inclusion criteria are supposed to be preset. Before the initiation of the study. According to the way presented here, these lines fit to the ''results'' section. 

Lines 228-233: Again, this part does not belong to Materials section. Here, you should only mention the methodology and nothing more. This part maybe fits better to the Discussion section, unless you rephrase.

Lines 381-387: That part is a result, it does not have to do with methodology.

Lines 596: There is no purpose on mentioning the type of the study, all the time. You mentioned it in the abstract and in the Materials and Methods section. That's enough.

Comments on the Quality of English Language

Moderate editing required.

Author Response

Reviewer 3

Dear authors.

I have carefully reviewed the current manuscript. It is a really interesting study in a common disorder in cats, however the presentation of the manuscript as a whole, is bad.

In the current form, the manuscript is not suitable for publication. The Materials and Methods section should be re-written. Many information mentioned in the Materials section should be in the Results section. Many of the Results are abruptly mentioned.

I have made some recommendations below; however you have to make a substantial progress in revising the manuscript, in order to be appropriate for publication.

We kindly appreciate the reviewers’ comments and think we will be able to cope with all of them, improving our manuscript and following all recommendations.       

Line 139: There is no need to mention the type of the study there (prospective, observational, cohort pilot study). That belongs to the Materials and Methods section, as you have already done.

We kindly appreciate the reviewer´s comments and, as suggested, we removed this part from the sentence.

Line 151: in this section, you should mention the number of the cats participated in the study

As suggested by the reviewer, we added the number of cats (line 151).

Line 153: ''had'' instead of ''were from''

We thank the reviewer for this comment and have replaced the words.

Line 153-167: The way you present the inclusion criteria is totally wrong and should be re-written. The inclusion criteria are supposed to be preset. Before the initiation of the study. According to the way presented here, these lines fit to the ''results'' section. 

Lines 228-233: Again, this part does not belong to Materials section. Here, you should only mention the methodology and nothing more. This part maybe fits better to the Discussion section, unless you rephrase.

Lines 381-387: That part is a result, it does not have to do with methodology.

We thank the reviewer for all these comments regarding to inclusion criteria and the neurorehabilitation protocol applied. As suggested, we have re-written this part (lines 151-163; lines 225-232; lenes 353-354; lines 374-378; line 434-435)

Lines 596: There is no purpose on mentioning the type of the study, all the time. You mentioned it in the abstract and in the Materials and Methods section. That's enough.

As suggested by the reviewer we removed the type of study throughout the manuscript.

Reviewer 4 Report (New Reviewer)

Comments and Suggestions for Authors

In this article, the authors conduct a prospective study with a population of 22 cats suffering from traumatic brachial plexus injuries to assess their response to a multimodal rehabilitation protocol (FES, kinesiotherapy, and treadmill exercises). Such pilot studies are essential for exploring new treatment modalities and understanding their effectiveness in specific conditions, such as brachial plexus injuries in cats. Early rehabilitation and functional electrical stimulation may play a crucial role in functional recovery after nerve injuries by aiming to enhance neuromuscular connection and promote neuronal plasticity.

However, despite the promising results obtained regarding the effectiveness of these interventions in the context of brachial plexus injuries in cats, it is important to note that these findings should be interpreted with caution. The study involves a limited number of subjects and requires broader, controlled studies to validate the findings. Nevertheless, due to the appropriate approach and development of the topic, the presentation of results, and ultimately the study design, this article meets the necessary quality standards for publication.

Author Response

Reviewer 4

In this article, the authors conduct a prospective study with a population of 22 cats suffering from traumatic brachial plexus injuries to assess their response to a multimodal rehabilitation protocol (FES, kinesiotherapy, and treadmill exercises). Such pilot studies are essential for exploring new treatment modalities and understanding their effectiveness in specific conditions, such as brachial plexus injuries in cats. Early rehabilitation and functional electrical stimulation may play a crucial role in functional recovery after nerve injuries by aiming to enhance neuromuscular connection and promote neuronal plasticity.

However, despite the promising results obtained regarding the effectiveness of these interventions in the context of brachial plexus injuries in cats, it is important to note that these findings should be interpreted with caution. The study involves a limited number of subjects and requires broader, controlled studies to validate the findings. Nevertheless, due to the appropriate approach and development of the topic, the presentation of results, and ultimately the study design, this article meets the necessary quality standards for publication.

We kindly appreciate the reviewer´s comment and taking all in consideration we added a sentence regarding the caution in data interpretation on line 788.

Round 2

Reviewer 3 Report (New Reviewer)

Comments and Suggestions for Authors

Dear authors.

As I have already mentioned, it is a really interesting study, however the first presentation was problematic (especially the Materials and Methods section).

The manuscript has been improved.

Good luck.

Comments on the Quality of English Language

Minor editing required.

Author Response

We kindly appreciate the reviewer’s comments and the authors have revised the manuscript, improved its quality and followed all recommendations.  

This manuscript is a resubmission of an earlier submission. The following is a list of the peer review reports and author responses from that submission.

Round 1

Reviewer 1 Report

Comments and Suggestions for Authors

The article takes a broad approach to the subject. In particular, the authors were unable to delineate or emphasize the topic. The results contain a significant amount of information, with an excess of tables/figures. The references could be better utilized by simplifying the number of citations.

Author Response

Reviewer 1

The article takes a broad approach to the subject. In particular, the authors were unable to delineate or emphasize the topic. The results contain a significant amount of information, with an excess of tables/figures. The references could be better utilized by simplifying the number of citations.

We kindly thank the reviewer for this comment. We have tried our best to re-write the introduction and the results section in order to emphasize the main topic and simplifying the number of citations. Also, we removed a figure from the results section.

Reviewer 2 Report

Comments and Suggestions for Authors

Interesting topic that I think is publishable if many corrections are made. Overall I think the study needs to be much more clear and succinct, how were the cats diagnosed, where was the lesion anatomically, what type of lesion was it (neuropraxia, axonotmesis, and neurotmesis), how this impacted the outcome, and it also needs to not overstate the ES as there was no control group to gauge the effect of the ES. Detailed comments are below.

There are a lot of authors, and the reviewer wonders if they all meet the guidelines below:

Authorship

MDPI follows the International Committee of Medical Journal Editors (ICMJE) guidelines which state that, in order to qualify for authorship of a manuscript, the following criteria should be observed:

  • Substantial contributions to the conception or design of the work; or the acquisition, analysis, or interpretation of data for the work; AND
  • Drafting the work or reviewing it critically for important intellectual content; AND
  • Final approval of the version to be published; AND
  • Agreement to be accountable for all aspects of the work in ensuring that questions related to the accuracy or integrity of any part of the work are appropriately investigated and resolved.

Those who contributed to the work but do not qualify for authorship should be listed in the acknowledgments. More detailed guidance on authorship is given by the International Council of Medical Journal Editors (ICMJE).

Any change to the author list should be approved by all authors including any who have been removed from the list. The corresponding author should act as a point of contact between the editor and the other authors and should keep co-authors informed and involve them in major decisions about the publication. We reserve the right to request confirmation that all authors meet the authorship conditions.

For more details about authorship please check MDPI ethics website.

It could also use editing by a native English speaker familiar with the content – there are numerous grammatical errors (way too many to list and/or correct).

The title “The role of early functional electrostimulation in partial traumatic brachial plexus injury cats: a pilot study” I think should be changed to “

The role of early rehabilitation and functional electrostimulation in partial traumatic brachial plexus injury cats: a pilot study as electrical stim was not the only thing used. The study does not show any increased rate of nerve recovery in cats with ES as there was no control group.

Lines 94-99 might expand on the success rate of these surgeries in people as it varies based on time from injury to surgery and the location damaged in the brachial plexus (for example nerve root avulsion).

Lines 171 – 192 The participants section could be clarified, did all cats had a confirmed PTBPI following MRI or CT or were some diagnosed only on clinical examination.

The authors describe the Seddon/Sunderland classifications in detail in the introduction. It would be good to describe the number of cats with neuropraxia, axonotmesis, and neurotmesis. How many were in each category? In addition, where the lesion was is not adequately described. For example was the lesion in a single root, multiple, were the patients categorized by the location of the nerve root based on clinical exam?

Lines 374 -382 - could you explain more about the distal depolarization. Is there evidence to suggest this would help the brachial plexus injury?

Line 494, with interferential current the pulse duration is not usually adjustable. Could you explain further how this was done?

“This technique used two different channels crossed each other at a 90â—¦ angle: one channel with 80–150 Hz, 0.5–1 mA, pulse duration 2–50 μs, 10 min; the other channel with 1–10 Hz, 0.5–1 mA, pulse duration 100–400 μs, 10 min [54].”

Was the pulse duration modulated in channel 1 from 2-50uS and in channel 2 from 100-400? This would be very unusual to modulate the pulse duration like this. Typically it is set at a single number within the range that was specified.

Table 2 why is the suprascapular, axillary, musculocutaneous not mentioned?  

Table 4 should provide the pulse duration used, not just the frequency of the FES

Also in Table 4 it would be useful to provide the laser dosage in Joules/cm2, not just Joules.

Table 5  - variance is not needed,

Table 6 – why did 20/22 cats have knuckling in the hindlimb? That seems like a very high percentage if it is just a brachial plexus injury without spinal trauma.

References that should be added:

Sawaya SG, Combet D, Chanoit G, Thiebault JJ, Levine D, Marcellin-Little DJ. Assessment of impulse duration thresholds for electrical stimulation of muscles (chronaxy) in dogs. Am J Vet Res. 2008;69(10):1305-1309. doi:10.2460/ajvr.69.10.1305

This study reports appropriate pulse durations for small animals by the individual muscle group. In a positive light – your choices for pulse duration do align with this paper.

Drum MG, Bockstahler B, Levine D, Marcellin-Little DJ. Feline rehabilitation. Vet Clin North Am Small Anim Pract. 2015;45(1):185-201. doi:10.1016/j.cvsm.2014.09.010

This paper discusses feline rehabilitation including NMES and would be a good introductory reference.

Marcellin-Little DJ, Levine D. Principles and application of range of motion and stretching in companion animals. Vet Clin North Am Small Anim Pract. 2015;45(1):57-72. doi:10.1016/j.cvsm.2014.09.004

Reference for end-feel (line 274)

Comments on the Quality of English Language

needs moderate editing for language

Author Response

Reviewer 2

Interesting topic that I think is publishable if many corrections are made. Overall, I think the study needs to be much clearer and succinct, how were the cats diagnosed, where was the lesion

anatomically, what type of lesion was it (neuropraxia, axonotmesis, and neurotmesis), how this impacted the outcome, and it also needs to not overstate the ES as there was no control group to gauge the effect of the ES. Detailed comments are below.

There are a lot of authors, and the reviewer wonders if they all meet the guidelines below.

It could also use editing by a native English speaker familiar with the content – there are numerous grammatical errors (way too many to list and/or correct).

  • The title “The role of early functional electrostimulation in partial traumatic brachial plexus injury cats: a pilot study” I think should be changed to “The role of early rehabilitation and functional electrostimulation in partial traumatic brachial plexus injury cats: a pilot study as electrical stim was not the only thing used. The study does not show any increased rate of nerve recovery in cats with ES as there was no control group.

We kindly appreciate the reviewer´s comment and following the suggestion, we have re-written the title.

  • Lines 94-99 might expand on the success rate of these surgeries in people as it varies based on time from injury to surgery and the location damaged in the brachial plexus (for example nerve root avulsion).

Thank you so much for this comment and meaning suggestion. We have re-written this paragraph, adding some of these important information (line 87-89).

  • Lines 171 – 192 The participants section could be clarified, did all cats had a confirmed PTBPI following MRI or CT or were some diagnosed only on clinical examination. The authors describe the Seddon/Sunderland classifications in detail in the introduction. It would be good to describe the number of cats with neuropraxia, axonotmesis, and neurotmesis. How many were in each category? In addition, where the lesion was is not adequately described. For example, was the lesion in a single root, multiple, were the patients categorized by the location of the nerve root based on clinical exam?

We kindly appreciate the reviewer´s comment and to clarify we re-wrote all the inclusion criteria, specifying that all cats have indeed performed MRI or CT (line 152-167).

However, regarding the description in each lesion category, we only know that cats had partial lesions compatible to axonotmesis or neurotmeses, located on the caudal portion. These was based on the neurologists reports and clinical examination, but the detail of where the lesion was, in a single or multiple roots, we cannot know and were not informed by the neurologist’s report, and neither by the MRI/CT reports. Most neurologists only made reference to the radial nerve lesion.

  • Lines 374 -382 - could you explain more about the distal depolarization. Is there evidence to suggest this would help the brachial plexus injury?

Thank you so much for this comment. Here we are talking about the pelvic limb. In cats that have brachial plexus injury but signs in the pelvic limb, it is likely they have spinal cord lesion. Thus, in those cases, the descending motor pathways and their information to the intumescence is decreased and consecutively to the nerve. ES is an attempt to increase depolarization, aiming to improve the proprioceptive deficits.

Regarding to the evidence of the use in brachial plexus injury, findings have been shown that brief continuous stimulation with 20 Hz accelerates axon outgrowth in rats or mice (Al-Majed et al. 2000), as animal models. Additionally, there is a possible effect in accelerating target reinnervation described in humans (and changes due to brain derived neurotrophic factor (BDNF) and its TrkB receptors upregulation (Gordon et al. 2010). Also, ES has been shown to have potential enhancing regeneration in different types of nerve injuries, including crush lesions, transection and long-distance injuries (Joved et al. 2021). Most studies that are performed in animals’ resort to a low-frequency ES, usually 20 Hz (Chiaramonte et al. 2023).

  • Line 494, with interferential current the pulse duration is not usually adjustable. Could you explain further how this was done? “This technique used two different channels crossed each other at a 90â—¦ angle: one channel with 80–150 Hz, 0.5–1 mA, pulse duration 2–50 μs, 10 min; the other channel with 1–10 Hz, 0.5–1 mA, pulse duration 100–400 μs, 10 min [54].” Was the pulse duration modulated in channel 1 from 2-50uS and in channel 2 from 100-400? This would be very unusual to modulate the pulse duration like this. Typically, it is set at a single number within the range that was specified.

We kindly appreciate these comments and, as suggested, we specify the pulse duration in the manuscript as well as re-written the paragraph (line 476-478). However, for better understanding, first it is used the two channels crossed at a 90º with the following parameters: one channel with 80–150 Hz, 0.5–1 mA, pulse duration 50 μs, 10 min; and then the two crossed channels with 1–10 Hz, 0.5–1 mA, pulse duration 159 μs, 10 min. Thus, a total of 20 minutes treatment.

  • Table 2 why is the suprascapular, axillary, musculocutaneous not

mentioned?

Thank you so much for this comment, we decided to mention only the caudal brachial plexus nerves given the lesion was always caudal. We re-wrote for better understanding in line 192 and in the title of the table.

  • Table 4 should provide the pulse duration used, not just the frequency of the FES. Also, in Table 4 it would be useful to provide the laser dosage in Joules/cm2, not just Joules.

We kindly thank the reviewer for this comment and added the pulse duration in table 4, as well as J/cm2.

  • Table 5 - variance is not needed

As suggested by the reviewer, we removed the variance from table 5.

Table 6 – why did 20/22 cats have knuckling in the hindlimb? That seems like a very high percentage if it is just a brachial plexus injury without spinal trauma.

We kindly appreciate the reviewer for this comment; however the 20/22 cats are referring to the absence of knuckling in the hindlimbs and only 2/22 cats had presence of knuckling in the hindlimbs, due to concurrent spinal cord injury.

References that should be added:

Sawaya SG, Combet D, Chanoit G, Thiebault JJ, Levine D, Marcellin-Little DJ. Assessment of impulse duration thresholds for electrical stimulation of muscles (chronaxy) in dogs. Am J Vet Res. 2008;69(10):1305-1309. doi:10.2460/ajvr.69.10.1305

This study reports appropriate pulse durations for small animals by the individual muscle group. In a positive light – your choices for pulse duration do align with this paper.

Drum MG, Bockstahler B, Levine D, Marcellin-Little DJ. Feline rehabilitation. Vet Clin North Am Small Anim Pract. 2015;45(1):185-201. doi:10.1016/j.cvsm.2014.09.010

This paper discusses feline rehabilitation including NMES and would be a good introductory reference.

Marcellin-Little DJ, Levine D. Principles and application of range of motion and stretching in companion animals. Vet Clin North Am Small Anim Pract. 2015;45(1):57-72. doi:10.1016/j.cvsm.2014.09.004

Reference for end-feel (line 274)

We kindly appreciate these bibliographic suggestions and have added them in the manuscript (line 806, 854 and 867).

Reviewer 3 Report

Comments and Suggestions for Authors

General comments:

The publication describes an interesting approach to a serious neurological problem. The method is described in detail and the results are accurately presented. Points of improvement are a more logical structure and a correct bundling of information in the introduction and discussion. There are also a number of language and grammatical errors and some sentences are very long. Therefore, the text does not read smoothly, especially in combination with the many abbreviations. I pointed out some language errors (indicated in red) but not all of them and I did not correct them.

Introduction:

the introduction is quite long and the content has no logical order. On the other hand, nothing is said about the dates in relation to the duration of recovery and results.

Line 76-93: The classification of nerve injuries is described in detail: is this relevant information? Most terms are not used further in the text.

Line 134-148: you do not mention in which species research has been performed.

Materials and methods:

Quite extensive description but interesting for those who want to apply the treatment.

Results:

Some grammatical errors.

Figure 11 is confusing: you want to illustrate an arthrodesis but there was also a surgical treatment of the radius and ulna. Either you choose a different illustration of a carpal arthrodesis or you mention the additional fractures (and when that treatment was performed) – to be honest: it is not the best example…

Discussion:

Again, there are several language errors and long sentences.

The division into paragraphs does not always seem to be done correctly. An example is the text section of line 662-681.

Line 694: to which study do you refer to?

Line 746-751: no phantom pain was seen as in humans, but can this be demonstrated in cats?

Comments on the Quality of English Language

There are also a number of language and grammatical errors and some sentences are very long. Therefore, the text does not read smoothly, especially in combination with the many abbreviations. I pointed out some language errors (indicated in red) but not all of them and I did not correct them.

Author Response

Reviewer 3

The publication describes an interesting approach to a serious neurological problem. The method is described in detail and the results are accurately presented. Points of improvement are a more logical structure and a correct bundling of information in the

introduction and discussion. There are also a number of language and grammatical errors and some sentences are very long. Therefore, the text does not read smoothly, especially in combination with the many abbreviations. I pointed out some language errors (indicated in red) but not all of them and I did not correct them.

  • Introduction: the introduction is quite long and the content has no logical order. On the other hand, nothing is said about the dates in relation to the duration of recovery and results.

We kindly thank the reviewer for this comment. We have tried our best to re-write the introduction and simplify for better understanding. Also, we have added the time regarding recovery (line 79 and 81).

  • Line 76-93: The classification of nerve injuries is described in detail: is this relevant information? Most terms are not used further in the text.

We appreciate the reviewer´s comment and followed your suggestion, removing this part from the introduction.

  • Line 134-148: you do not mention in which species research has been performed.

As suggested by the reviewer, we added in this paragraph the information about the species. In these studies, the animal models were rats.

  • Materials and methods: Quite extensive description but interesting for those who want to apply the treatment.

We kindly thank the reviewer for this comment. The importance of these methods, as a repeatable treatment, was one of our aims and that is why the extensive description.

Results:

Some grammatical errors.

  • Figure 11 is confusing: you want to illustrate an arthrodesis but there was also a surgical treatment of the radius and ulna. Either you choose a different illustration of a carpal arthrodesis or you mention the additional fractures (and when that treatment was performed) – to be honest: it is not the best example...

We kindly appreciate the reviewer´s comment and as suggested we removed this figure.

Discussion:

  • Again, there are several language errors and long sentences. The division into paragraphs does not always seem to be done correctly. An example is the text section of line 662-681.

We kindly thank the reviewer and, as suggested, we tried our best to improve language and grammar errors, as well as the division in paragraphs for a better understanding. Also, we have tried to take some of the abbreviations of the manuscript.

  • Line 694: to which study do you refer to?

Thank you so much for this comment. In fact, this sentence is no study but our own opinion regarding the use of the splint to avoid carpal contractures secondary to brachial plexus injury. This could be used for some hours a day. The immobilization has detrimental effects in the number of fibers with reduction in the muscle regeneration rate and secondary nociceptive pain due to the joint contractures, which in case of brachial plexus injury manifests in carpus contracture. Also, in human medicine the use of modalities with a splint is described to help in contractures of the carpus (Charamote et al. 2023).

However, to avoid misinterpretation we choose to remove this sentence.

  • Line 746-751: no phantom pain was seen as in humans, but can this be demonstrated in cats?

We kindly appreciate this comment and question. Yes, in our study it was not seen any signs of possible phantom pain. According to Menchetti et al. (2017), there is evidence of a phantom complex in amputated animals, reporting signs, such as reduction in activity and participation in the family life or everyday activities. Also, in Menchetti et al. (2022) there is one specific study about cats that express manifestations of pain before and after amputation, which is described by the owners (nearly in 41% of the cases), and very often associated to a stressful behavior. As pointed out by the reviewer, we thought that it would beneficial to add this last study to our citations. Thank you once again for the possibility to improve our study (line 716).

Therefore, we didn´t see any of these signs in our cats, during the 4 years of follow-up, as mentioned in the manuscript.

Round 2

Reviewer 1 Report

Comments and Suggestions for Authors

The authors met most of the considerations.